# OOD-MAML: Meta-Learning for Few-Shot Out-of-Distribution Detection and Classification

**Taewon Jeong**    **Heeyoung Kim**
Department of Industrial and Systems Engineering
KAIST
Daejeon 34141, Republic of Korea
`{papilion89,heeyoungkim}@kaist.ac.kr`

## Abstract

We propose a few-shot learning method for detecting out-of-distribution (OOD) samples from classes that are unseen during training while classifying samples from seen classes using only a few labeled examples. For detecting unseen classes while generalizing to new samples of known classes, we synthesize fake samples, i.e., OOD samples, but that resemble in-distribution samples, and use them along with real samples. Our approach is based on an extension of model-agnostic meta learning (MAML) and is denoted as OOD-MAML, which not only learns a model initialization but also the initial fake samples across tasks. The learned initial fake samples can be used to quickly adapt to new tasks to form task-specific fake samples with only one or a few gradient update steps using MAML. For testing, OOD-MAML converts a $K$-shot $N$-way classification task into $N$ sub-tasks of $K$-shot OOD detection with respect to each class. The joint analysis of $N$ sub-tasks facilitates simultaneous classification and OOD detection and, furthermore, offers an advantage, in that it does not require re-training when the number of classes for a test task differs from that for training tasks; it is sufficient to simply assume as many sub-tasks as the number of classes for the test task. We also demonstrate the effective performance of OOD-MAML over benchmark datasets.

## 1   Introduction

Deep neural networks (DNNs) have demonstrated excellent performances in many machine learning tasks such as speech recognition [20], object detection [8], and image classification [10]. However, they usually require large amounts of training data to perform well. When the amount of training data is small, DNNs often provide low levels of performance. This is a critical issue, because a large number of problems in the real world are confronted with the lack of training data. For example, the drug discovery problem involves the prediction of whether a molecule increases the pharmaceutical activity [1]. However, often only a small amount of data relating to molecules are available, which makes it difficult to use DNNs for this task. The learning problem under the circumstance wherein only a few labeled examples are available is referred to as *few-shot learning*, and it has attracted significant interest recently [24, 3, 21, 7]. "$K$-shot $N$-way" classification is one of the major tasks that few-shot learning deals with, wherein a small number of samples $K$ (such as 1,5, or 10) per each of $N$ classes are available for training to classify a new sample into one of the $N$ classes. Several methods, e.g., siamese network [13] and prototypical network (PN) [22], were proposed for $K$-shot $N$-way problems.

In general, in few-shot classification algorithms, it is assumed that the training and test data are drawn from the same distribution, and the algorithm requires a test sample to be classified into one of the known classes encountered during training. However, in many real-world applications, it is unreasonable to assume the same distribution for the training and test data [5, 23, 4]. If the test data are drawn from a different distribution, it is more desirable to have classification algorithms that

not only classify the test samples drawn from the same distribution of training data, but also detect out-of-distribution (OOD) samples drawn from an unknown distribution. Humans can easily do this even in a few-shot scenario. For example, let us suppose that a child learns the alphabets 'A' and 'B' from a few examples. This child would then not only discriminate between 'A' and 'B', but also can say "I did not learn it" on seeing 'C' and 'D'. In supervised learning settings, methods have been proposed for detecting OOD samples, and the majority of these methods are based on uncertainty quantification (UQ) for predictions [15, 18]. However, the algorithms in previous studies are not designed for few-shot settings.

In this paper, we propose a method for detecting OOD samples from unseen classes during training while classifying samples from known classes under few-shot settings. There are two major challenges in the problem of few-shot OOD detection and classification: (1) a lack of training data required for learning the distribution of the data from known classes and (2) the absence of OOD samples during training.

To address the first challenge, we use meta-learning, which is a general paradigm for few-shot learning. The objective of meta-learning is to learn a learning strategy to learn quickly on new tasks. In general, meta-learning algorithms involve two core processes: learning the transfer of knowledge across tasks and rapid adaptation to a new task. In this study, we use model-agnostic meta-learning (MAML) [7], which is a popular gradient-based meta-learning algorithm. The objective of MAML is to find good initial parameters of a model (e.g., DNN), such that updating the initial parameters via one or a few gradient steps can result in a model that provides a good performance for a new task. More details about MAML are presented in Section2.2.

Although meta-learning is a good approach to few-shot classification, it does not address the second challenge. In the absence of OOD examples from unknown classes, meta-learning algorithms, including MAML, would generate trivial classifiers that predict all the examples as in-distribution examples. To address this issue, we propose the synthesis of OOD examples from unknown classes, which are then used along with in-distribution examples to learn a classifier. We were inspired by MetaGAN [26], which uses adversarial samples generated from GAN to help few-shot classifiers learn a sharper decision boundary. Similarly, we synthesize adversarial samples to represent an OOD class during training. However, instead of using GAN, we generate adversarial samples via gradient updating for special meta-parameters, called *fake-sample* parameters. This generation strategy allows us to avoid the difficulty of training the GAN. The proposed method is called OOD-MAML.

To facilitate the OOD detection, OOD-MAML is trained to adapt quickly for OOD tasks with respect to a single class. In meta-testing phase, given $K$-shot $N$-way samples, we construct $N$ sub-tasks of $K$-shot OOD detection with respect to each class. Then $N$ classifiers are adapted to each sub-task with meta-knowledge. By merging the results of $N$ OOD detection tasks, we can implement OOD detection and $K$-shot $N$-way classification simultaneously. This approach has an advantage, in that it does not require re-training when the number of classes for the test tasks are changed. The code for OOD-MAML is available at `https://github.com/twj-KAIST/OOD-MAML`.

## 2 Background

### 2.1 Task formulation in general meta-learning algorithms

We first discuss the task formulation considered in general meta-learning algorithms. We deal with two types of meta-sets, one is for meta-training and the other is for meta-testing, which are denoted by $D_{meta-train}$ and $D_{meta-test}$, respectively. Each of $D_{meta-train}$ and $D_{meta-test}$ contains multiple datasets, each of which is divided into a training set $D_{train}$ and a test set $D_{test}$ as in the typical classification task. Thus, we can rewrite $D_{meta-train} = \{(D_{train}^i, D_{test}^i)_{i=1}^{N_{meta-train}}\}$ and $D_{meta-test} = \{(D_{train}^j, D_{test}^j)_{j=1}^{N_{meta-test}}\}$, where $N_{meta-train}$ and $N_{meta-test}$ are the numbers of classification tasks in $D_{meta-train}$ and $D_{meta-test}$, respectively, and $(D_{train}^i, D_{test}^i)$ denote the training and test sets for the $i$th task, $T_i$. In particular, on considering the $K$-shot $N$-way classification task, $D_{train}^i$ contains $K$ examples from each of $N$ classes of $T_i$. That is, $D_{train}^i$ consists of a total of $N \times K$ examples. Similarly, $D_{test}^i$ contains some examples, each of which is assumed to be drawn from one of the $N$ classes considered during the training. Generally, it is assumed that $D_{meta-train}$ and $D_{meta-test}$ are exclusive, i.e., $D_{meta-train}$ and $D_{meta-test}$ do not share the same

task. Therefore, in meta-learning methods, the ability of generalizing or adapting to new tasks is trained in the meta-training phase, and the trained ability is evaluated in the meta-testing phase.

## 2.2 Model-Agnostic Meta-Learning

MAML [7] is a popular optimization-based meta-learning approach that learns the initial model parameters, which result in fast learning on new tasks through gradient-based optimization. More specifically, MAML takes a DNN, $f_\theta$ with meta-parameter $\theta$, as the base model, and learns the $\theta$ that allows fast adaptation to new tasks when used as the initial parameters. Given $D^i_{train}$ for task $T_i$, MAML implements the adaptation of $\theta$ to task-specific parameters $\theta^i_{adapt}$ via gradient updates of $\theta$ with respect to $L_{T_i}$, which is the loss for $T_i$. On assuming a single gradient step, the update equation is given by

$$\theta^i_{adapt} = \theta - \alpha \nabla_\theta L_{T_i}(f_\theta(D^i_{train})),$$

where $\alpha > 0$ is the adaptation learning rate. In a few-shot classification, the cross-entropy function is commonly used as a loss function. The adaptation of $\theta$ is then evaluated on $D_{test}$. In MAML, $\theta$ is trained by optimizing the performance of $\theta^i_{adapt}$ with respect to $\theta$ for all the tasks. That is, $\theta$ is trained to be $\theta_{meta-opt} = \arg\min_\theta \sum_{T_i \sim P(T)} L_{T_i}(f_{\theta^i_{adapt}}(D^i_{test}))$, where $P(T)$ is the distribution of tasks. MAML can be interpreted from the perspective of fine-tuning, such that the objective of MAML is to learn a good initial parameter of the base model after one or a few gradient update steps using only a few labeled samples. MAML updates $\theta$ across tasks via stochastic gradient descent with the meta learning rate $\beta$:

$$\theta \leftarrow \theta - \beta \nabla_\theta \sum_{T_i \sim P(T)} L_{T_i}(f_{\theta^i_{adapt}}(D^i_{test})).$$

## 2.3 Related works on OOD detection

OOD detection using DNNs is known to be a challenging problem because DNNs can produce incorrect high-confidence predictions for OOD samples. For example, DNNs can label random noise static to a particular object class with over 99% confidence [19]. This tendency of DNNs interrupts the OOD detection. In order to address this issue, several methods based on UQ have been proposed and have demonstrated successful performances. Softmax scores of pretrained DNNs are used for UQ based on the observation that DNNs tend to assign higher softmax scores for in-distribution (seen class) samples than OOD (unseen class) samples [11]. Out-of-DIstribution detector for Neural network(ODIN) improved this method by applying additional techniques of temperature scaling and adversarial perturbation [18]. UQ based on Mahalanobis distance (denoted as MAH) are studied for OOD detection under the assumption that pre-trained features of softmax DNNs follow the class-conditional Gaussian mixture distribution [15]. After the posterior mean and covariance with respect to the in-distribution class are estimated, MAH calculates the Mahalanobis distance of a test input with respect to the posterior mean for each class. MAH also used the adversarial perturbation and demonstrated the state-of-the-art performance in OOD detection.

## 3 Few-Shot OOD Detection with MAML (OOD-MAML)

We present the training and testing procedures used in our meta-learning method for few-shot OOD detection and classification. Let us consider the example of the learning process of children in Section 1, and let us suppose that a child learns 'A' and 'B' in a step-by-step manner. In the process of learning 'A,' this child attempts to understand two concepts: what is 'A' and what is not 'A.' This learning process is similar to that used for learning 'B.' We could interpret the learning process for each alphabet as an OOD detection with respect to the learned alphabet. When the child then sees 'C,' the child would combine the conclusions of 'it is not A' and 'it is not B.' Similarly, for the alphabet 'B,' the conclusions of 'it is not A' and 'it is B' would be integrated. In this manner, a child can perform OOD-detection and classification simultaneously even in a few-shot scenario. Similarly, our method can perform OOD detection and classification simultaneously using the classifier that is trained to detect OOD examples for a given known class. We refer to our method as OOD-MAML as we use the adaptation rule of MAML.

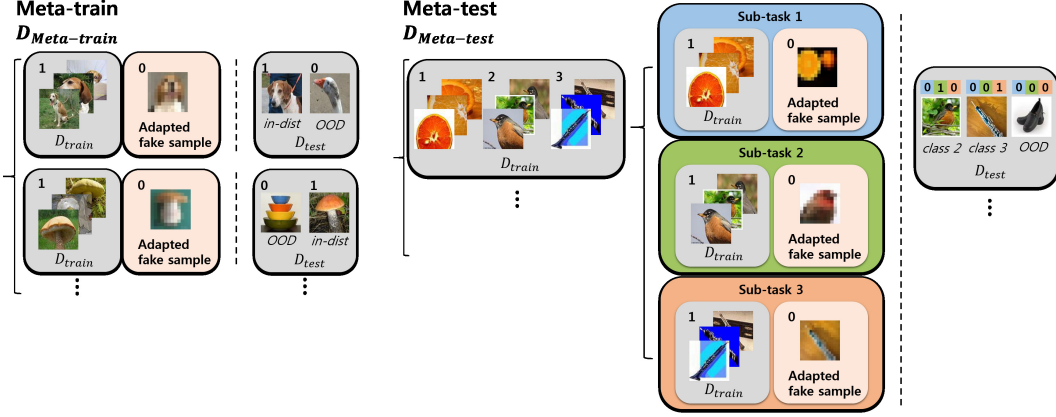

Figure 1: Example of OOD-MAML setup

## 3.1 Task formulation in OOD-MAML

Like other general meta-learning algorithms, we deal with $D_{meta-train}$ and $D_{meta-test}$ for meta-training and meta-testing, respectively, but construct them differently. In our work, we allow test examples for each task to be drawn not only from seen classes but also from unseen classes. To facilitate few-shot OOD detection, we construct $D_{train} \in D_{meta-train}$ differently from that assumed in general meta-learning algorithms. We construct $D_{train} \in D_{meta-train}$ to contain $K$ examples of *one* known class. That is, for each task $T_i$, $D_{train}$ is constructed by $D_{train}^i = \{\mathbf{x}_1^i, \ldots, \mathbf{x}_K^i\}$, where each example $\mathbf{x}_k^i$ is assigned the same label of 1. To facilitate the OOD detection, we artificially generate OOD examples, which are used to train the base model to learn a sharper decision boundary. The OOD examples are generated as the adversarial samples for $D_{train}^i$. During the test time, we evaluate the meta-trained model for both $K$-shot $N$-way classification and OOD detection. To do this, in $D_{meta-test}$, we assume $D_{train} \in D_{meta-test}$ have a $K$-shot $N$-way setting. Given task $T_j$, we denote $D_{train}^j = \{\{\mathbf{x}_{1k}^j\}_{1 \le k \le K}, \{\mathbf{x}_{2k}^j\}_{1 \le k \le K}, \ldots, \{\mathbf{x}_{Nk}^j\}_{1 \le k \le K}\}$, where $\mathbf{x}_{nk}^j$ denote $k$ samples of the $n$th class. In the perspective of OOD detection, examples of $N$ classes are in-distribution examples, but our meta learner is trained in a situation wherein $D_{train}^i$ contains samples of only a single class. In order to match the situation in the training and testing, we split $T_j$ into multiple sub-tasks $\{T_{jn}\}_{n=1,2,\ldots,N}$, where the $n$th class is the only seen class in $T_{jn}$. Given sub-task $T_{jn}$, we define $D_{train}^{jn} = \{\mathbf{x}_{nk}^j\}_{1 \le k \le K}$, such that only samples of the $n$th class belong to the seen class for $T_{jn}$, and we obtain the adapted parameters for $T_{jn}$. We set $D_{test}^j$ to contain samples of several classes including $N$ classes in $D_{train}^j$ for each task $T_j$. The ability of OOD-MAML to perform OOD detection and classification for $T_j$ is evaluated on $D_{test}^j$ from the combined results of the OOD detection tasks of all $T_{jn}$. The proposed meta-learning procedure is illustrated in Figure 1.

## 3.2 Meta-training procedure for OOD-MAML

Let us consider a base model $f_\theta$ parameterized by a DNN with the meta-parameter $\theta$. From the notations in Section 3.1, we define $L_{\theta;T_i}^{in}$ as the cross-entropy loss for $D_{train}^i$: $L_{\theta;T_i}^{in} = -\frac{1}{K} \sum_{k=1}^{K} \log f_\theta(\mathbf{x}_k^i)$. If we adapt $\theta$ with a gradient update using MAML (i.e., $\theta^i = \theta - \alpha \nabla_\theta L_{\theta;T_i}^{in}$), the adapted base parameter would be biased and the adapted base model would become a trivial classifier (i.e., $f_{\theta_b^i}(x) = 1$ for an arbitrary input $x$) because all the elements of $D_{train}^i$ belong to the same class.

To address this issue, we introduce a fake-sample parameter vector $\theta_{fake} = (\theta_{fake,1}, \ldots \theta_{fake,M})$, which is another meta-parameter. It plays the role of $M$ initial fake samples across tasks and quickly adapts to new tasks to form task-specific fake samples, i.e., unseen class samples with respect to $D_{train}^i$. When $\mathbf{x}_k^i \in R^d$, we set each $\theta_{fake,m} \in R^d$. We treat $\theta_{fake}$ as initial fake samples and assign them the label 0. We then define an additional loss $L_{\theta,T_i}^{out}(\theta_{fake}) = -\frac{1}{M} \sum_{m=1}^{M} \log\left(1 - f_\theta(\theta_{fake,m})\right)$, which is the cross entropy loss for $\theta_{fake}$. Then we denote $L_{\theta;T_i}(D_{train}^i, \theta_{fake}) = L_{\theta;T_i}^{in} + L_{\theta;T_i}^{out}(\theta_{fake})$. This loss is used for gradient updates with respect to

$\theta$. Using the adaptation learning rate $\alpha > 0$, we first update $\theta$ to $\theta^i$ for task $T_i$ as follows:

$$\theta^i = \theta - \alpha \nabla_\theta L_{\theta;T_i}(D^i_{train}, \theta_{fake}). \tag{1}$$

The next step involves adapting $\theta_{fake}$ to task $T_i$. Note that without adapting $\theta_{fake}$, $f_{\theta^i}$ sets $\theta_{fake}$ to be the representatives of OOD class for all $T_i$. Then, $\theta_{fake}$ has the same value for all $T_i$, which indicates that $f_{\theta^i}$ can capture task-agnostic OOD concepts only. To detect OOD samples more precisely, task-specific OOD concepts should be learned, and thus it is required to adapt $\theta_{fake}$. The issue to be resolved is how to adapt $\theta_{fake}$. Here, we are motivated from the idea of MetaGAN [26], where it was argued that adversarial samples generated using GAN, even with an imperfect generator, help an adapted classifier to learn a sharper decision boundary. However, instead of using GAN, we generate adversarial samples via gradient update. Note that the purpose of adapting $\theta_{fake}$ is to generate helpful OOD class samples with respect to $T_i$, wherein the helpful OOD samples mean the adversarial samples of OOD class such that the classifier with $\theta_i$ in Eq.(1) predicts them wrongly. We implement this via adversarial gradient updating of $\theta_{fake}$ with respect to $L_{\theta^i;T_i}(D^i_{train}, \theta_{fake})$, and can avoid training GAN, which involves the difficulty such as vanishing gradient [25] and mode collapsing [17].

However, the amount of the gradient of $L_{\theta^i;T_i}(D^i_{train}, \theta_{fake})$ for $\theta_{fake}$ can be small, and thus standard adversarial gradient updating can lead to small perturbation of $\theta_{fake}$. This is not desirable because we expect the adapted $\theta_{fake}$ to provide different information (i.e., task-specific OOD information) from $\theta_{fake}$. To resolve this issue, we combine the sign-gradient and meta-SGD [16] for adapting $\theta_{fake}$. The adapted fake samples for task $T_i$, $\theta^i_{fake}$, are computed as follows:

$$\theta^i_{fake} = \theta_{fake} - \beta_{fake} \odot sign(-\nabla_{\theta_{fake}} L_{\theta^i;T_i}(D^i_{train}, \theta_{fake})), \tag{2}$$

where $\beta_{fake} > 0$ is another meta-parameter to be learned, but unlike $\theta$ and $\theta_{fake}$, it is not adapted for each $T_i$. $\odot$ denotes the element-wise product. Note that $sign(-\nabla_{\theta_{fake}} L_{\theta^i;T_i}(D^i_{train}, \theta_{fake}))$ in Eq.(2) provides the direction of adversarial updating, and $\beta_{fake}$ determines how much to update with the direction. In meta-SGD [16], there is no restriction for $\beta_{fake}$, but we restrict it to be positive because maintaining the adversarial direction can be helpful to generate the adversarial samples with respect to $f_{\theta^i}$[9].

To reflect the adversarial input $\theta^i_{fake}$ into the base model, we return to updating $\theta$ with the gradient of $L_{\theta^i;T_i}(D^i_{train}, (\theta_{fake}, \theta^i_{fake}))$, where $(\theta_{fake}, \theta^i_{fake})$ means the concatenation of $\theta_{fake}$ and $\theta^i_{fake}$. Note that compared with Eq.(1), $\theta^i_{fake}$ is now added to the representatives of OOD class. The final adapted base parameter $\theta^i_{adapt}$ is obtained as follows:

$$\theta^i_{adapt} = \theta - \alpha \nabla_\theta L_{\theta^i;T_i}(D^i_{train}, (\theta_{fake}, \theta^i_{fake})). \tag{3}$$

With the final adapted classifier $f_{\theta^i_{adapt}}$, we run meta-optimization across tasks via stochastic gradient descent, similar to MAML. When $D^i_{test} = \{(\mathbf{x}^i_1, y^i_1), (\mathbf{x}^i_2, y^i_2), \ldots, (\mathbf{x}^i_Q, y^i_Q)\}$ are given for each $T_i$, where $y^i_q = 1$ if $\mathbf{x}^i_q$ is a seen class sample and $y^i_q = 0$ otherwise, we train all meta-parameters $(\theta, \theta_{fake}, \beta_{fake})$ as follows:

$$(\theta, \theta_{fake}, \beta_{fake}) \leftarrow (\theta, \theta_{fake}, \beta_{fake}) - \gamma \nabla_{(\theta, \theta_{fake}, \beta_{fake})} \sum_{T_i \sim P(T)} L(D^i_{test}), \tag{4}$$

where $L(D^i_{test}) = -\frac{1}{Q} \sum_{q=1}^Q y^i_q \log p^i_q + (1 - y^i_q) \log(1 - p^i_q)$, $p^i_q = f_{\theta^i_{adapt}}(\mathbf{x}^i_q)$, and $\gamma > 0$ is the meta-learning rate.

## 3.3 Meta-testing procedure for OOD-MAML

Given sub-task $T_{jn}$ from $D^{jn}_{train}$, we obtain $\theta^{jn}_{adapt}$ in the same manner as in Eqs.(1),(2), and (3). Now, we validate the adaptation to the samples in $D^j_{test}$. Given $x \in D^j_{test}$, we concatenate the adaptation results for $x$ from $T_{jn}$s, i.e., $p^j(x) = [f_{\theta^{j1}_{adapt}}(x), \ldots, f_{\theta^{jN}_{adapt}}(x)]$, where $p^j(x)$ denotes the $K$-shot $N$-way results for $T_j$. Note that $f_{\theta^{jn}_{adapt}}(\cdot)$ are binary classifiers, and the label 0 can be assigned if $f_{\theta^{jn}_{adapt}}(\cdot) < \lambda$, where $\lambda$ is a threshold, while the label 1 is assigned otherwise, in the

test phase. The threshold $\lambda$ can be determined based on some criteria such as the true positive ratio (TPR), or simply set to 0.5 as a default value for binary classification. In our experiments in Section 4, we set the threshold at 95 or 98 % TPR. More details on the determination of $\lambda$ based on TPR are described in Section 4.1. If all the $N$ elements in $p^j(x)$ are less than $\lambda$, we assign the unseen class (out-of-distribution) for $x$. Otherwise, we assign the maximum index of $p^j(x)$ as the class for $x$ among $N$ classes, i.e., the index of the assigned class for $x$ is equal to $\mathrm{argmax}_{1 \leq n \leq N} f_{\theta_{adapt}^{jn}}(x)$.

Figure 1 depicts the OOD-MAML procedure discussed in Sections 3.2 and 3.3.

## 4    Experiments

We run the experiments for few-shot OOD detection and classification tasks with OOD-MAML. In the meta-training phase, we set the 5-shot data of one class in $D_{train}$ and set 50 samples in $D_{test}$, where 25 samples are drawn from seen classes (i.e., classes encountered in $D_{train}$) and the remaining 25 samples are drawn from unseen classes. In the meta-testing phase, we set 5-shot 5-way data for $D_{train}$, and we set $D_{test}$ to contain 50 samples of the same setting of $D_{test}$ in the meta-training phase. Under these settings, we evaluated the performance of OOD-MAML by implementing OOD detection and classification in experiments, and compared the obtained results with the performances of several OOD detection methods. First, we considered ODIN and MAH. To apply these methods, pre-trained softmax classifiers are required. For pre-trained classifiers, we constructed an MAML model for 5-shot 5-way classification. To avoid any confusion with $D_{meta-train}$ in OOD-MAML, we denote the meta-training data set for this MAML model as $D_{meta-train}^{MAML}$. $D_{train}$ and $D_{test}$ in $D_{meta-train}^{MAML}$ consist of 5-shot 5-way datasets. After training this MAML model with $D_{meta-train}^{MAML}$ (see Section 2.2), the base model is adapted to $D_{train}$ in the meta-testing phase. We applied ODIN and MAH to this adapted model. Moreover, we apply ODIN with PN [22], such that ODIN's techniques are used for softmax over distances to the prototypes in PN. Furthermore, we considered ($N$+1) classes with MAML, $N$ in-distribution classes and 1 OOD class, with the same set of $D_{meta-train}$. Here, we consider two cases: (1)($N$+1) classes with MAML without fake images and (2) ($N$+1) classes with MAML with $\theta_{fake}$ and $\theta_{fake}^i$ (denote as ($N$+1)-MAML and ($N$+1)-MAML*, respectively). We ran experiments on Omniglot [14], CIFAR-FS [2], and *mini*ImageNet [24], which are popular benchmark datasets used for few-shot learning.

### 4.1    Evaluation criteria

ODIN and MAH are score-based methods. They detect OOD samples based on whether the score of a sample is higher than a fixed threshold. In both cases, the threshold is selected such that the true positive rate, i.e., the ratio of positive (in-distribution) samples correctly classified as positive samples, is sufficiently high (both works set it to 95% ). In our comparison, we also set the threshold based on the true positive rate (TPR) for $D_{train}$ of $D_{meta-test}$. To measure the OOD detection performance of ODIN and MAH, we used the true negative rate (TNR) at $\alpha$% TPR. This measure is interpreted as the probability that negative (OOD) samples are classified correctly as negative when the TPR is $\alpha$%. It is computed using $TNR = TN/(TN + FP)$, where $TN$ and $FP$ denote the numbers of true negatives and false positives, respectively. A perfect detector has a 1.0 TNR value. In addition to this, we measure the detection accuracy, which is the ratio of correctly discriminated in- and out-of-distribution samples among $D_{test}$. We set equal numbers of positive and negative samples in the test set (25 of 50 as in-distribution samples, and the remaining 25 as OOD samples). Thus, the detection accuracy is not biased to in- or out-of- distribution. It is also measured at $\alpha$% TPR. We also set the threshold $\lambda$ for OOD-MAML discussed in Section 3.3 at $\alpha$% TPR. More specifically, we first meta-trained OOD-MAML and chose 1000 different OOD-detection tasks from $D_{meta-train}$, for each of which we adapted our base classifier and then calculated in-distribution probability for each of positive instances (i.e., in-distribution samples) in the test data. Then, based on all calculated in-distribution probabilities, we determined $\alpha$% TPR threshold. Because ($N$+1)-MAML and ($N$+1)-MAML* are not score-based classifiers, their performance do not depend on $\alpha$. We also compared the classification accuracy for the $K$-shot $N$-way classification of OOD-MAML and MAML.

### 4.2    Details of the neural network architecture

Our base model in both OOD-MAML and MAML has a convolution neural network (CNN) architecture, which has four modules for Omniglot and CIFAR-FS and five modules for *mini*ImageNet. Each module consists of $3 \times 3$ convolutions and 64 filters for Omniglot and CIFAR-FS, and 32 filters for *mini*ImageNet, which are followed by batch normalization [12], the exponential linear units (ELU) activation function [6], and max-pooling with $2 \times 2$ stride and padding. The dimension of the final

Table 1: OOD detection results

| | ODIN -MAML | ODIN -PN | MAH -MAML | $(N+1)$ -MAML | $(N+1)$ -MAML* | OOD -MAML($M$=3) | OOD -MAML($M$=5) |
|---|---|---|---|---|---|---|---|
| **Omniglot** | | | | | | | |
| detect.acc $\alpha=95$ | 0.8744 (0.0512) | 0.8977 (0.0441) | 0.8712 (0.0481) | 0.9142 (0.0392) | 0.9524 (0.0341) | 0.9712 (0.0296) | 0.9701 (0.0288) |
| detect.acc $\alpha=98$ | 0.8912 (0.0331) | 0.9122 (0.0287) | 0.8320 (0.0785) | | | 0.9838 (0.0225) | 0.9833 (0.0214) |
| TNR $\alpha=95$ | 0.6942 (0.1142) | 0.7122 (0.0533) | 0.7288 (0.0821) | 0.8722 (0.0730) | 0.9201 (0.0633) | 0.9924 (0.0224) | 0.9918 (0.0225) |
| TNR $\alpha=98$ | 0.7124 (0.0988) | 0.7369 (0.0629) | 0.7544 (0.0233) | | | 0.9831 (0.0342) | 0.9839 (0.0359) |
| **CIFAR-FS** | | | | | | | |
| detect.acc $\alpha=95$ | 0.5811 (0.1022) | 0.5933 (0.1113) | 0.5601 (0.0891) | 0.5035 (0.1299) | 0.5531 (0.1021) | 0.6752 (0.0738) | 0.6612 (0.0813) |
| detect.acc $\alpha=98$ | 0.6129 (0.1132) | 0.6039 (0.1285) | 0.5458 (0.0671) | | | 0.6590 (0.0719) | 0.6594 (0.0725) |
| TNR $\alpha=95$ | 0.2311 (0.1291) | 0.1592 (0.1422) | 0.2999 (0.1239) | 0.1051 (0.1833) | 0.2017 (0.0945) | 0.5492 (0.1250) | 0.5512 (0.1311) |
| TNR $\alpha=98$ | 0.2401 (0.1087) | 0.1439 (0.1027) | 0.1862 (0.1244) | | | 0.4317 (0.1287) | 0.4198 (0.1249) |
| ***mini*ImageNet** | | | | | | | |
| detect.acc $\alpha=95$ | 0.5124 (0.0742) | 0.5491 (0.0981) | 0.5111 (0.1124) | 0.5019 (0.0712) | 0.5422 (0.1101) | 0.6207 (0.0736) | 0.6199 (0.0744) |
| detect.acc $\alpha=98$ | 0.5641 (0.0411) | 0.5669 (0.1003) | 0.5229 (0.1174) | | | 0.6125 (0.0750) | 0.6024 (0.0688) |
| TNR $\alpha=95$ | 0.1211 (0.1899) | 0.1829 (0.1042) | 0.1429 (0.1366) | 0.0749 (0.0822) | 0.1009 (0.1033) | 0.6770 (0.1181) | 0.6613 (0.1203) |
| TNR $\alpha=98$ | 0.1659 (0.1426) | 0.1942 (0.0819) | 0.1944 (0.1209) | | | 0.4902 (0.1310) | 0.4891 (0.1287) |

Table 2: Classification accuracy results. $M$=3 for OOD-MAML

| Method | Omniglot | CIFAR -FS | *mini* Imagenet | Method | Omniglot | CIFAR -FS | *mini* Imagenet |
|---|---|---|---|---|---|---|---|
| MAML ($K$=5,$N$=5) | 0.9911 (0.0371) | 0.7084 (0.1230) | 0.5926 (0.1086) | OOD-MAML ($K$=5,$N$=3) | 0.9996 (0.0054) | 0.7220 (0.1121) | 0.6322 (0.0989) |
| OOD-MAML ($K$=5,$N$=5) | 0.9989 (0.0071) | 0.7158 (0.1129) | 0.6044 (0.1098) | OOD-MAML ($K$=5,$N$=7) | 0.9894 (0.0287) | 0.6964 (0.1139) | 0.5822 (0.1319) |

CNN layer is 256, 256, and 288 for Omniglot, CIFAR-FS, and *mini*ImageNet, respectively. The last layer is fed into a softmax, where its dimension is 2 in the case of OOD-MAML and 5 in the case of MAML. As all the real images in the three considered datasets are bounded in $[0, 1]$ for each dimension, we re-scaled the fake images (i.e., $\theta_{fake}$ and $\theta^i_{fake}$) using the sigmoid transformation to obtain the same domain. We ran the experiments while changing the number of fake images ($M$). Details about hyperparameters are described in Supplementary material.

### 4.3 Experiment results

Table 1 reports the few-shot OOD detection performance of OOD-MAML with $\alpha\%$ TPR threshold and the considered competing methods. It lists the average and standard deviation of the TNR and detection accuracy over 1000 different tasks. We can observe that OOD examples were detected more effectively using OOD-MAML than others. In particular, OOD-MAML demonstrated a significant improvement in TNR. We also report OOD detection results of OOD-MAML with $\lambda$ being simply set as 0.5 in Supplementary material.

Next, we evaluated the few-shot classification performance of OOD-MAML and compared it with that of MAML. Table 2 lists the average and standard deviation of the 5-shot 5-way classification accuracy over 1000 different tasks. We measured the classification accuracy over the examples of the five (known) classes, while excluding the OOD examples. As listed in Table 2, OOD-MAML performed slightly better or similarly to MAML. We may thus conclude that OOD-MAML is more

advantageous than MAML because OOD-MAML can also perform OOD detection while achieving a comparable classification performance. Moreover, we could perform classification tasks with various numbers of classes ($N$) using OOD-MAML without re-training it over different values of $N$. This is another advantage of OOD-MAML. As shown in Table 2, the performances of OOD-MAML were similar for tasks with different $N$. This shows that OOD-MAML is robust to changes in the number of classes.

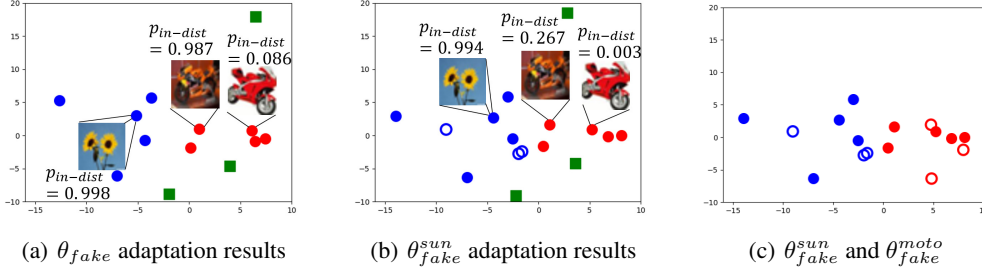

(a) $\theta_{fake}$ adaptation results      (b) $\theta_{fake}^{sun}$ adaptation results      (c) $\theta_{fake}^{sun}$ and $\theta_{fake}^{moto}$

Figure 2: Feature map results from OOD task $T_{sun}$, where 'sunflower' is in-distribution class and 'motorcycle' is OOD class. In (a) and (b), $\theta^{sun}$ (Eq.(1)) and $\theta_{adapt}^{sun}$ (Eq.(3)) are used as the final adapted base parameter, respectively. Blue circles: features of in-distribution images. Red circles: features of OOD images. Green squares: features of $\theta_{fake}$. Blue rings: features of $\theta_{fake}^{sun}$. Red rings: features of $\theta_{fake}^{moto}$, which were generated from OOD task $T_{moto}$, where 'motorcycle' is in-distribution class.

### 4.4 Effects of adapted fake samples

In order to investigate the effects of adapted fake samples in Eq.(2) on the constructed decision boundary for OOD tasks, we compared the behaviors of classifiers adapted by Eq.(1) and Eq.(3) after running the meta-training of OOD-MAML. Here, we denote the former and latter classifier as $\theta_{fake}$-classifier and $(\theta_{fake} + \theta_{fake}^i)$-classifier (i.e., the classifier of OOD-MAML), respectively. Note that $\theta_{fake}$ is fixed for all tasks, and thus $\theta_{fake}$-classifier is adapted by the representatives of task-agnostic OOD-samples, while $(\theta_{fake} + \theta_{fake}^i)$-classifier is adapted by task-specific OOD-samples (Eq.(2)). We argue that task-specific OOD samples lead to a better decision boundary for OOD tasks than task-agnostic OOD samples.

After the meta-training of OOD-MAML with benchmark dataset (CIFAR-FS), we compared the extracted features of in-distribution samples, OOD samples, $\theta_{fake}$, and $\theta_{fake}^i$ on both $\theta_{fake}$-classifier and $(\theta_{fake} + \theta_{fake}^i)$-classifier. We first chose one OOD task, $T_{sun}$, wherein 'sunflower' is in-distribution class. Then we take the 'motorcycle' as OOD class. Figure 2 visualizes the features for both classifier with respect to $T_{sun}$, where the features were extracted from the outputs of the final layer before softmax. Principle component analysis was applied to depict the features into 2-dim spaces. Figures 2(a) and 2(b) depict the features on $\theta_{fake}$-classifier and $(\theta_{fake} + \theta_{fake}^{sun})$-classifier for $T_{sun}$, respectively. In both figures, blue and red circles represent the features of in- and out-of distribution images from the same dataset, respectively; the green squares represent the features of $\theta_{fake}$ for both classifiers.

In Figure 2(a), $\theta_{fake}$-classifier constructs the decision boundary based on in-distribution samples (blue circles) and $\theta_{fake}$ (green squares) assigned as OOD class. In the feature space, in-distribution samples and $\theta_{fake}$ are located far away from each other, which results in a loose decision boundary for OOD detection. The loose decision boundary subsequently led to wrong predictions for some OOD samples whose features are located near those of some in-distribution samples (see Figure 2(a)). In Figure 2(b), blue rings represent the features of $\theta_{fake}^{sun}$ on $(\theta_{fake} + \theta_{fake}^{sun})$-classifier. Note that Eq.(2) leads the features of $\theta_{fake}^{sun}$ to be located near the in-distribution samples, and $(\theta_{fake} + \theta_{fake}^{sun})$-classifier is adapted to assign $\theta_{fake}^{sun}$(and $\theta_{fake}$) to OOD-class. This leads to a tighter decision boundary than that in Figure 2(a), which subsequently leads to a higher detection accuracy. For example, the wrongly

predicted OOD image with the estimated in-distribution probability as 0.987 in Figure 2(a) was predicted correctly with the estimated in-distribution probability as 0.267 in Figure.2(b).

Next, we generated $\theta_{fake}^{moto}$, adapted fake samples from $T_{moto}$, the OOD tasks with 'motorcycle' as in-distribution class. The red rings in Figure 2(c) represent the features of $\theta_{fake}^{moto}$ on $(\theta_{fake} + \theta_{fake}^{sun})$-classifier. We can find that the features of $\theta_{fake}^{moto}$ are located near the features of motorcycle images (red circles), while the features of $\theta_{fake}^{sun}$ (blue rings) are located near the features of sunflower images (blue circles). This shows that our process to generate adapted fake images (Eq.(2)) successfully produced different outputs adaptively depending on the tasks.

## 5  Conclusion

We proposed OOD-MAML, which is a meta-learning method used for implementing $K$-shot $N$-way classification and OOD detection simultaneously. In OOD-MAML, we introduced two types of meta-parameters: one is related to the base model as in the case of MAML, and the other type, *fake-sample* parameters, plays the role of generating OOD samples. Based on the hypothesis regarding MetaGAN that adversarial samples work as additional training signals to the base model as well as make the decision boundary sharper, we adapt the fake sample parameters as adversarial samples via the gradient update with adversarial loss. Our future work could be focused on developing a more efficient training method for OOD-MAML.

## Broader Impact

OOD-MAML can help humans to detect abnormal behaviors quickly and take appropriate actions in a variety of real-world problems, including production system monitoring, preventive maintenance, fraud detection, health condition monitoring, and disease surveillance. OOD-MAML can contribute to the machine learning community by providing a new perspective to OOD detection as a new, supervised, approach. Previous methods for OOD detection have focused on an unsupervised learning framework to construct the decision boundary of in-distribution samples. However, this approach generally requires a huge amount of in-distribution samples and also can suffer from model uncertainty. Instead, we take a supervised learning framework by introducing an adapted classifier, which is evaluated not only with in-distribution samples, but also with OOD samples in the meta-training phase.

## Acknowledgments

This work was partly supported by Samsung Electronics Co.,Ltd. and the National Research Foundation of Korea (NRF) grant funded by the Korea Government (MSIT) (No.2018R1C1B6004511). The authors would like to thank the reviewers for providing valuable comments. The authors would also like to thank Youngmin Lee for active discussions during the research.

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
