[Supplementary Material]

# A    Pseudo-code of OOD-MAML

---

**Algorithm 1** OOD-MAML with $K$-shot training samples

---

**Require:** $p(T)$: distribution over tasks of OOD detection
**Require:** $\alpha, \gamma > 0$: hyper-parameter

 1: randomly initialize $\theta, \theta_{fake}$ and $\beta_{fake}$
 2: **while**  not done **do**
 3:    Sample batch of $T_i \sim p(T)$
 4:    **for** all $T_i$ **do**
 5:       Sample $K$ data points $D^i_{train} = \{\mathbf{x}^i_1, \mathbf{x}^i_2, \ldots, \mathbf{x}^i_K\}$
 6:       First update $\theta$ with Eq.(1):
           $\theta^i = \theta - \alpha \nabla_\theta L_{\theta; T_i}(D^i_{train}, \theta_{fake})$
 7:       Update $\theta_{fake}$ with Eq.(2):
           $\theta^i_{fake} = \theta_{fake} - \beta_{fake} \odot sign(-\nabla_{\theta_{fake}} L_{\theta^i; T_i}(D^i_{train}, \theta_{fake}))$
 8:       Compute the final adapted base parameter $\theta^i_{adapt}$, with Eq.(3):
           $\theta^i_{adapt} = \theta - \alpha \nabla_\theta L_{\theta^i; T_i}(D^i_{train}, (\theta_{fake}, \theta^i_{fake}))$
 9:       Sample data points
           $D^i_{test} = \{(\mathbf{x}^i_1, y^i_1), (\mathbf{x}^i_2, y^i_2), \ldots, (\mathbf{x}^i_Q, y^i_Q)\}$ from $T_i$ for meta-update
10:    **end for**
11:    update $\boldsymbol{\theta} \leftarrow \boldsymbol{\theta} - \gamma \nabla_{\boldsymbol{\theta}} \sum_{T_i \sim P(T)} L(D^i_{test})$
       where $\boldsymbol{\theta} = (\theta_b, \theta_{fake}, \beta_{fake})$
12: **end while**

---

# B    Datasets

**Omniglot** (Lake et al., 2015) is a dataset of handwritten characters and contains 20 examples of 1623 characters. Omniglot is the most commonly used dataset in few-shot learning, and its images are resized to $28 \times 28$ (Finn et al., 2017; Santoro et al., 2016; Snell et al., 2017; Sung et al., 2018; Koch et al., 2015). As in other studies, we randomly select 1200 characters for meta-training and use the remaining for meta-testing.

***mini*ImageNet** is a sub-dataset of ImageNet (Russakovsky et al., 2015). It contains a total of 60K images of 100 different classes, each of which comprises 600 RGB images. Ravi and Larochelle (2016) presented the protocol for ***mini*ImageNet** as per which all the images are downsampled to $84 \times 84$ and are divided into 64 classes for meta-training, 16 classes for meta-validation, and 20 for meta-testing. We followed this protocol but did not use the meta-validation set.

**CIFAR-FS** is a sub-dataset of CIFAR-100. It contains 600 RGB images per each of the 100 classes. Bertinetto et al. (2018) resized all the images to $32 \times 32$ and divided this dataset into 64 classes for meta-training, 16 classes for meta-validation, and 20 classes for meta-testing. We used datasets of

64 and 20 classes for the meta-training and meta-testing, respectively.

# C   OOD detection results of OOD-MAML with $\lambda = 0.5$

Table 1: OOD detection results

| | ODIN -MAML | ODIN -PN | MAH -MAML | $(N{+}1)$ -MAML | $(N{+}1)$ -MAML* | OOD -MAML($M{=}3$) | OOD -MAML($M{=}5$) |
|---|---|---|---|---|---|---|---|
| | Omniglot | | | | | | |
| detect.acc $\alpha = 95$ | 0.8744 (0.0512) | 0.8977 (0.0441) | 0.8712 (0.0481) | 0.9142 (0.0392) | 0.9524 (0.0341) | 0.9683 (0.0339) | 0.9788 (0.0381) |
| detect.acc $\alpha = 98$ | 0.8912 (0.0331) | 0.9122 (0.0287) | 0.8320 (0.0785) | | | | |
| TNR $\alpha = 95$ | 0.6942 (0.1142) | 0.7122 (0.0533) | 0.7288 (0.0821) | 0.8722 (0.0730) | 0.9201 (0.0633) | 0.9380 (0.0674) | 0.9429 (0.0639) |
| TNR $\alpha = 98$ | 0.7124 (0.0988) | 0.7369 (0.0629) | 0.7544 (0.0233) | | | | |
| | CIFAR-FS | | | | | | |
| detect.acc $\alpha = 95$ | 0.5811 (0.1022) | 0.5933 (0.1113) | 0.5601 (0.0891) | 0.5035 (0.1299) | 0.5531 (0.1021) | 0.6637 (0.0737) | 0.6519 (0.0819) |
| detect.acc $\alpha = 98$ | 0.6129 (0.1132) | 0.6039 (0.1285) | 0.5458 (0.0671) | | | | |
| TNR $\alpha = 95$ | 0.2311 (0.1291) | 0.1592 (0.1422) | 0.2999 (0.1239) | 0.1051 (0.1833) | 0.2017 (0.0945) | 0.4558 (0.1295) | 0.4624 (0.1281) |
| TNR $\alpha = 98$ | 0.2401 (0.1087) | 0.1439 (0.1027) | 0.1862 (0.1244) | | | | |
| | $mini$ImageNet | | | | | | |
| detect.acc $\alpha = 95$ | 0.5124 (0.0742) | 0.5491 (0.0981) | 0.5111 (0.1124) | 0.5019 (0.0712) | 0.5422 (0.1101) | 0.6218 (0.1099) | 0.6129 (0.1184) |
| detect.acc $\alpha = 98$ | 0.5641 (0.0411) | 0.5669 (0.1003) | 0.5229 (0.1174) | | | | |
| TNR $\alpha = 95$ | 0.1211 (0.1899) | 0.1829 (0.1042) | 0.1429 (0.1366) | 0.0749 (0.0822) | 0.1009 (0.1033) | 0.6386 (0.1204) | 0.6372 (0.1196) |
| TNR $\alpha = 98$ | 0.1659 (0.1426) | 0.1942 (0.0819) | 0.1944 (0.1209) | | | | |

# D   Hyperparameter settings

We set the learning rates in the adaptation process as $\alpha = 0.1$. In the meta optimizing process (Eq.(4)), we used the Adam optimizer (Kingma and Ba, 2015) with the learning rate $\gamma$=0.001 and meta-batch size of 4.

In the meta-training phase, the parameters were updated via one gradient step using Eqs.(1) and and three gradient steps using Eq.(2) and Eq.(3). In order to reduce the computation cost, we used

the first-order approximation for updating $\theta_{fake}$.

In the meta-testing phase, the parameters were updated via three gradient step using Eqs.(1), and five gradient steps using Eq.(2) and Eq.(3).

# E    Further experiments

As previously mentioned, OOD-MAML involves two types of meta-parameters $\theta_{fake}$ and $\theta$, which are learned interactively. We performed additional experiments to check whether $\theta_{fake}$ is really learned interactively with $\theta$ across tasks and also check whether the adapted fake samples $\theta_{fake}^i$ really help the classifier to better perform OOD detection.

We first constructed $D_{meta-test}$ in the same manner as that used for $D_{meta-train}$, i.e., $D_{train}^i$ contained examples of one class, and $D_{test}^i$ contained examples of multiple classes including the class of $D_{train}^i$. Then from the trained OOD-MAML for each data source, we changed the adversarial sample generating process in two ways. The first one is to replace the adapted fake samples by random fake samples $u \sim U[0,1]^d$. The resulting method is denoted as random-OOD-MAML. For random-OOD-MAML, we changed the loss for the gradient update in Eq.(3) by replacing $\theta_{fake}^i$ with $u$ and then adapt the parameters of the base model via a gradient update with respect to the changed loss.

The other way in which we changed the adversarial sample generating process was to replace the trained $\theta_{fake}$ by arbitrary random initial inputs and follow the adversarial adaptation with respect to these random initials. The resulting method is denoted as random-($ini$)-OOD-MAML. We compared the OOD detection accuracy for $D_{meta-test}$ using OOD-MAML, random-OOD-MAML, and random-($ini$)-OOD-MAML over 1000 tasks as shown in Table 2. Note that random-OOD-MAML and random-($ini$)-OOD-MAML share the initial parameter $\theta$ of the base model that has already been meta-trained. The results thus obtained show that OOD-MAML outperformed the other two. This shows that $\theta_{fake}$ and $\theta$ were interactively meta-trained effectively, and thus, the adapted fake samples provide more useful information for OOD detection than random OOD samples. Moreover, the superior performance of random-($ini$)-OOD-MAML over that of random-OOD-MAML shows that the application of the adversarial adaptation strategy in Eq.(2) is reasonable for improving

the performance.

Table 2: OOD detection accuracy of OOD-MAML, random-OOD-MAML, and random-($ini$)-OOD-MAML

| Method | Omniglot | CIFAR-FS | $mini$ Imagenet |
|---|---|---|---|
| OOD-MAML $M$=3 | 0.9812 (0.0128) | 0.7921 (0.0939) | 0.7012 (0.1081) |
| random-($ini$)-OOD-MAML, $M$=3 | 0.9764 (0.0122) | 0.7422 (0.1042) | 0.6739 (0.1012) |
| random-OOD-MAML, $M$=3 | 0.9233 (0.0548) | 0.6402 (0.1011) | 0.6115 (0.1129) |

# F    Comparison between OOD-MAML and MetaGAN

In this section, we discuss the commonalities and difference between OOD-MAML and MetaGAN. Both methods attempt to generate adversarial samples. In MetaGAN, the adversarial samples provide a sharper decision boundary for the adapted classifier with respect to $N$ classes. MetaGAN forces the classifier to learn how to classify among $N$ classes and also how to classify the real and fake samples. In order to implement this, the classifier must extract correct features for $N$ known classes and the OOD class (for fake samples). The use of correct features make the decision boundary stricter. The OOD-MAML is also intended to have this effect and adapted fake samples induce a tight decision boundary for seen classes. This is desirable because $D_{train}^i$ only contains seen class samples, and it is thus highly possible for the classifier to be biased towards seen classes after the adaptation, i.e., classifier has a broad decision boundary for seen classes. This is the most important issue that is required to be solved for the OOD detection task. In our work, this can be solved by using adapted fake samples.

One difference between OOD-MAML and MetaGAN is the method of generating adversarial samples. OOD-MAML generates adversarial samples by updating the fake-sample parameter via gradient descent, while MetaGAN uses GAN. Herein, we claim that our work is more efficient than MetaGAN in terms of the training. GAN requires a large amount of parameters, and MetaGAN makes use of neural networks having a ResNet-like architecture that involves as many parameters as over 100 times the input sample dimension. In contrast, the dimension of $\theta_{fake}$ is the same as that

of the input samples, which is significantly less than that of the parameter of GAN. Moreover, GAN presents the risk of significant information loss. MetaGAN comprises the use of the conditional GAN, which takes its input as the vector of the representatives of $D_{train}^i$ concatenated with random noise. MetaGAN introduces the instance-encoder module, which extracts features for each example in $D_{train}^i$. Therefore, after extracting the features, MetaGAN treats the average pooled vector of all the features as the representative of $D_{train}^i$. Here, if $N$ (number of classes) is larger, a severe loss of information about $D_{train}^i$ would occur, because it is difficult for one vector to contain the information of $N$ complex manifolds. In contrast, OOD-MAML results in less information loss than MetaGAN because it generates adversarial samples with respect to just one class.

Second, OOD-MAML and MetaGAN have different objectives in optimizing the meta-training. OOD-MAML is designed to optimize the same objective across tasks to train both $\theta$ and $\theta_{fake}$ (see Eq.(4)). In contrast, MetaGAN uses the adversarial meta-training objective, and thus, $f_\theta$ and GAN are optimized according to different objectives. Moreover, in the $K$-shot $N$-way classification problem, MetaGAN assigns adversarial samples to the label $(N+1)$, and thus, the base model $f_\theta$ is designed to be the output of $(N+1)$ logits. In MetaGAN, it can be interpreted that $\theta$ is optimized for the $K$-shot $N$-way problem across tasks, while the parameters for GAN are optimized to mimic an arbitrary $D_{train}^i$. In contrast, in OOD-MAML, $\theta$ and $\theta_{fake}$ are interactively trained to minimize the same loss across tasks and collaboratively updated in each adaptation phase. In this way, OOD-MAML generates adversarial samples that are helpful for OOD detection, in contrast to GAN generating arbitrary adversarial samples.

Finally, OOD-MAML has the advantage of flexibility in terms of task changes, while MetaGAN does not. When the target tasks are changed from $K$-shot $N$-way to $K$-shot $M$-way, MetaGAN is required to re-train the model because the outputs of the base model depend on the number of classes, which affects the training of the GAN (Note that for $K$-shot $N$-way, MetaGAN assigns the $N+1$th label to adversarial samples). Not only MetaGAN but also general meta-learning methods require re-training in this situation (Finn et al., 2017; Santoro et al., 2016). In contrast, OOD-MAML does not require re-training; only the number of sub-tasks is required to be changed from $N$ to $M$ because this change does not affect the training in our method.