[Reviews · NeurIPS 2020]

Review 1

Summary and Contributions: The submission explores simultaneous classification and out-of-distribution (OOD) detection in a few-shot setting. The underlying framework for the few-shot setting is MAML with binary classification meta-tasks. For OOD detection, the paper uses generated examples of OOD samples per class. MetaGAN [1] has proposed to use a GAN to generate such contrastive samples, but the current work uses an adversarial perturbation method instead based on the FGSM [2], with the sample initialisation being meta-learned. Experiments indicate that the proposed method might perform better than more naive MAML instantiations. A small case-study indicates that the adversarial perturbation per class is significantly useful for OOD detection in the proposed framework. [1] MetaGAN: An Adversarial Approach to Few-Shot Learning, Zhang et al., 2018 [2] Explaining and Harnessing Adversarial Examples, Goodfellow et al., 2015

Strengths: I find the idea of meta-learned initialisations for adversarial samples interesting, and potentially inspirational for other works. To my knowledge, this is quite novel.

Weaknesses: My comments are mostly to do with empirical methodology; see below.

Correctness: In Table 1, baseline methods are thresholded at a 95% TPR, while the proposed method and its variants are claimed to be threshold-agnostic: From section 3.3 it appears that the threshold is manually set to be 0.5, so they are not really threshold agnostic. It feels likely to me that there might be situations where picking thresholds with different criteria for comparative methods might lead to an unfair assessment. I’d recommend picking an OOD-detection threshold (on the maximum softmax values for class 1 across all tasks, for example) also at 95% TPR for a more even comparison. The experiment in Section 4.4 feels a bit anecdotal due to the particular example studied. Appendix D studies the effect of the adversarial adaptation, and while the text says random-(ini)-OOD outperforms random-OOD, the table seems to show the opposite trend (a typo perhaps?), which would indicate the adversarial adaptation did not help. In any case, aggregate numbers in the actual setting might be nice, with the meta-learned initialisations instead of random initialisations. It might also be useful to visualise the fake samples learned by the meta-training process, as well as the adversarially-perturbed ones. Do the authors suspect a particular reason for why performance goes down upon increasing the number of negative samples (M = 3 to 5) in Table 1 for miniImagenet (and to some small extent, for CIFAR-FS)? How were hyper-parameters picked for the various methods? From the Appendix it appears that validation sets were not used. While these are probably true, I’d be cautious about claiming them without some evidence: L45: "the algorithms in previous studies would not perform well under few-shot settings.” L61:"meta-learning algorithms, including MAML, would generate trivial classifiers that predict all the examples as in-distribution examples.” ---------------------- Post-rebuttal -------------------- Thanks for the responses to my comments. I'm happy with the updates.

Clarity: While the writing could probably be improved to make the method clearer, it wasn’t particularly difficult for me to follow. Here are some typos/suggestions: L31: “K-show” -> “K-shot” L142: “examples” -> “example” L173: “If then,” -> “If so,”/“Then” L267: “few-show”->”few-shot”

Relation to Prior Work: To my knowledge, prior work is adequately discussed

Reproducibility: Yes

Additional Feedback:


Review 2

Summary and Contributions: This work introduces a novel and challenging setting: few-shot learning for open-set. The authors propose an optimization-based method along with a data generation strategy to handle this task.

Strengths: This work introduces a challenging few-shot setting, which is rarely explored in previous works. The method introduced in this work is optimization-based and the data generation strategy can actually help to improve the performance for this task.

Weaknesses: 1. Terminology ambiguous. The "detection" in computer vision-related tasks means to localize and classify one instance of a specific class. 2. The author claims the GAN is difficult to optimize and use adversarial gradient updating. It's not clear why the proposed idea is better than GAN. How to avoid the issue of the vanishing gradient and model collapsing of their proposed method is less presented in the method section. 3. Can you provide some visualization of the fake example for better understanding and analysis?

Correctness: Correct

Clarity: The method section should be improved with more intuitive explanation and analysis.

Relation to Prior Work: This work is also related to the open-set problem in the classification task, the related works shoud be reported and compared with.

Reproducibility: Yes

Additional Feedback: The feedback has solved my concerns. The related work of open-set problem should be reported in the final version. I will keep my rating.


Review 3

Summary and Contributions: This paper studies OOD detection in the few-shot setting. It first formulates the few-shot OOD detection problem based on the classical few-shot setting. Then this paper proposes OOD-MAML to meta-train and meta-test the model. The main idea is to involve adversarial samples to train a sharp classifier which can help to identify the OOD samples. This paper also demonstrates the effective performance of OOD-MAML over benchmark datasets.

Strengths: - This paper discuss the few-shot OOD detection problem which is interesting. The authors rigorously formulate the problem based on the classical few-shot learning setting. - This paper proposes a MAML-like algorithm, OOD-MAML, to solve the problem. The main contribution is introduce adversarial samples (as meta-parameters) to help train the classifier, which is novel. This paper also carefully designed the gradient descent steps to update the meta-parameters. - The experiments are sufficient and show the effectiveness of OOD-MAML algorithm.

Weaknesses: The baseline in the experiment is MAML which is not designed for OOD detection, so I think it is not fair enough. Do you have other baselines? For example, many-shot OOD detection algorithms?

Correctness: Yes. I haven't seen anything wrong so far.

Clarity: Yes. One suggestion: The notations used in section 3 are a little bit too complex and could be simplified.

Relation to Prior Work: Yes. This work is related to MAML and OOD detection, which are both discussed in section 2.

Reproducibility: Yes

Additional Feedback:


Review 4

Summary and Contributions: Presents a meta-learning algorithm that meta-trains for detecting out of distribution samples. At meta-test time, OOD-MAML can do both N-way classification and OOD detection. The method is a modification of MAML. Meta-train: - Each task is essentially a binary classification problem where in distribution samples are examples from the same class (labeled 1) and out of distribution samples are generated adversarially (labeled 0) - To get the adv examples, they start with theta_fake, a meta-learned vector of adv examples. For each task, they directly gradient update theta_fake to fool the model. theta_fake is updated using sign of gradient, and has a meta-learned inner learning rate. - IIUC, they then need to update the model again to reflect the updated adv examples. Meta-test - The N-way K-shot problem is turned into N binary classification tasks again, essentially giving N binary classifiers, one for each class - Looking at the output of each of the N classifiers, if the output is < 0.5 for all classes then they say it is OOD - Otherwise, choose the class corresponding to the highest (maximum) output among the classifiers Experiments - OOD examples are generated by choosing from unseen classes - Compare against OOD detection baselines combined with MAML - OOD-MAML appears to better detect OOD examples, and has comparable accuracies for in distribution data

Strengths: - Method seems novel as far as I'm aware, in particular the ideas of turning few shot classification problem into few-shot in/out binary classification, and adapting the adversarial example directly in the inner loop - Evaluation appears to show benefits in detecting OOD examples, without hurting acc in distribution - Work is relevant to the community

Weaknesses: - Though the method is applied in a different setting (adding in OOD detection), the actual technical deviations from MAML and MetaGAN are relatively minor - It is not clear that the difference from MetaGAN (lack of a generator for adv examples) are justified or better, aside from being easier to train, and no experimental comparison is made to that approach. Post author response: The response is satisfactory, and my concerns were relatively minor to begin with. No change to score (7).

Correctness: The empirical methodology seems correct.

Clarity: The notation is somewhat cumbersome, many mathematical symbols stretch on for too long decreasing readability (D_{meta-train}^{MAML}, for example). Minor: too many equations are in-lined (perhaps for space) and decrease readability, e.g. L168. Figures and tables often lack captions explaining what is going on.

Relation to Prior Work: - Clearly discusses relation to other OOD methods - No discussion about prior meta-learning + OOD work, or comparison to them. However, I am not familiar enough with that area to say whether it is necessary. - Some discussion to other meta-learning work, particularly MAML and MetaGAN.

Reproducibility: Yes

Additional Feedback:

[Author Response · NeurIPS 2020]

We deeply appreciate the reviewers' careful comments. We hope all concerns can be resolved through our clarifications.

**<Reviewer 1>**

Q: I'd recommend picking an OOD detection threshould at 95% TPR for a more even comparison.

A: Thank you for your great suggestion. Previously we set the threshold at 0.5 as a default value for binary classification.
Following your suggestion, we re-performed the experiments by setting the threshold at 95% TPR. More specifically,
we first meta-trained OOD-MAML and chose 1000 different OOD-detection tasks from $D_{meta-train}$, for each of
which we adapted our base classifier and then calculated in-dist probability for each of positive instances (i.e., in-dist
samples) in the test data. Finally, based on all calculated in-dist probabilities, we picked 95% TPR threshold. In
this way, we obtained the threshold as larger than 0.5, which allowed a tighter decision boundary for in-dist samples
(0.9892 (Omniglot), 0.6183 (CIFAR-FS), 0.5255 (*mini*Imagenet)). With the new threshold, we could ensure more even
comparison, and even could improve the performance. The following table compares the new (first row) and previous
(second row) results. We will incorporate these new results in the final version.

|  | Omniglot | | CIFAR-FS | | *mini*ImageNet | |
|---|---|---|---|---|---|---|
|  | detect.acc | TNR | detect.acc | TNR | detect.acc | TNR |
| OOD-MAML ($M$=3, threshold at 95% TPR) | 0.9712 (0.0297) | 0.9924 (0.0224) | 0.6752 (0.0738) | 0.5491 (0.1250) | 0.6207 (0.0753) | 0.6770 (0.1182) |
| OOD-MAML ($M$=3, threshold=0.5) | 0.9683 (0.0339) | 0.9380 (0.0674) | 0.6637 (0.0737) | 0.4558 (0.1295) | 0.6218 (0.1099) | 0.6386 (0.1204) |

13
Q: The table in Appendix D shows the opposite trend (a typo perhaps?) of the text.
A: We are sorry that it is a typo: the name of the second and third rows should be switched, i.e., the second row is for
"random-(*ini*)-OOD-MAML" and the third row is for "random-OOD-MAML." We will fix this typo in the final version.
Q: The reason for the decreased performance upon increasing the number of negative samples ($M$=3 to 5).
A: It is possible that overfitting led to decreased performance in testing phase, as more parameters are used when $M$=5.
By the way, we found that there is a typo for *mini*ImageNet results in Table 1: TNR of OOD-MAML with $M$=5 should
be 0.6372 (0.1196). We will fix the type in the final version.
Q: How were the hyper-parameters chosen?
A: For OOD-MAML, we heuristically chose the hyper-parameters by evaluating meta-training loss. For other methods,
we chose them according to the settings reported in the MAML paper. We will clarify this in the final version.
Q: It should be more cautious to claim L45, L61 without some evidence.
A: We understand your concern. In the final version, we will modify L45 as "the algorithms in previous studies are not
designed for few-shot settings." For L61, we actually checked this behavior (i.e. MAML generates trivial classifiers for
OOD detection) by running experiments. We will add these experimental results in Appendix in the final version.

**<Reviewer 2>**

Q: How to avoid the issue of the vanishing gradient and mode collapsing of the proposed method is less presented.
A: We used sign-gradient of adversarial loss, which provides the direction for adversarial learning, and meta-SGD,
which provides the amount of perturbation (L185-189). By using them in combination for adapting $\theta_{fake}$, we found
that both vanishing gradient and mode collapsing issues (see Figure 2(c): different adaptation results) could be avoided.
We will discuss this in more detail in the method section in the final version.
Q: Related work of the open-set problem should be reported and compared with.
A: Surely, we will discuss the related work of open-set studies (e.g., [1,2]) in the final version.
[1] Boult, Terrance E., et al. (2019) "Learning and the unknown: Surveying steps toward open world recognition."
[2] Sehwag, Vikash, et al. (2019) "Analyzing the robustness of open-world machine learning."

**<Reviewer 3>**

Q: MAML, not designed for OOD detection, is not fair enough for comparison. Do you have any other baselines, e.g.,
many-shot OOD detection algorithms?
A: We agree that direct comparison with MAML is not fair enough because it is not designed for OOD detection;
our comparison with MAML is rather to show that we effectively extend MAML for ODD detection problems. In
Table 1, we have ODIN and MAH as other baselines of OOD detection methods. These methods require a pre-trained
classifier from many-shot training data in general and can be considered as many-shot OOD detection algorithms. For
fair comparison, we used an adapted classifier of MAML as the pre-trained classifier for ODIN and MAH.

**<Reviewer 4>**

Q: Need more clear discussion and comparison to Meta-GAN (lack of a generator for adv examples).
A: Fundamentally, OOD-MAML and MetaGAN have different objectives in meta-training phase. In MetaGAN, GAN
is (meta-) trained to generate adversarial samples across all tasks, while meta parameters for initial base model are
(meta-) trained to classify the instances in test data set after adaptation. Thus, the parameters for GAN and base model
are trained with different objectives. In contrast, in OOD-MAML, all meta parameters share the same objective (Eq.(4)),
and thus $\theta$ and $\theta_{fake}$ are interactively trained to minimize the same loss across tasks and collaboratively updated in
each adaptation phase. Thus, OOD-MAML generates adversarial samples that are helpful for OOD-detection. We will
clarify this in the final version. We will also compare MetaGAN and OOD-MAML via experiments.

[Meta-Review · NeurIPS 2020]

This paper presents a method for performing meta-learning and OOD detection. The reviewers agreed that this paper meets the bar for acceptance. For the camera ready, the authors are encouraged to address the reviewer's feedback that was discussed in the author response, address the other points of feedback (e.g. visualizing the fake samples), and carefully read through the paper for typos.